# Image-Guided Brachytherapy for Rectal Cancer: Reviewing the Past Two Decades of Clinical Investigation

**DOI:** 10.3390/cancers14194846

**Published:** 2022-10-04

**Authors:** Te Vuong, Aurelie Garant, Veronique Vendrely, Remi Nout, André-Guy Martin, Shirin A. Enger, Ervin Podgorsak, Belal Moftah, Slobodan Devic

**Affiliations:** 1Radiation Oncology Department, Jewish General Hospital, McGill University, Montreal, QC H3T 1E2, Canada; 2Department of Radiation Oncology, UT Southwestern Medical Center, Dallas, TX 75390, USA; 3CHU Bordeaux, Department of Radiation Oncology, F-33000 Bordeaux, France; 4BRIC (BoRdeaux Institute of onCology), UMR1312, INSERM, University of Bordeaux, F-33000 Bordeaux, France; 5Department of Radiotherapy, Erasmus MC Cancer Institute, University Medical Center Rotterdam, 3015 GD Rotterdam, The Netherlands; 6Service de Radio-Oncologie, CHU de Québec, Université Laval, Québec, QC G1V 4G2, Canada; 7Medical Physics Unit, Oncology Department, McGill University, Montreal, QC H3A 0G4, Canada; 8King Faisal Specialist Hospital & Research Centre, MBC 03, PO Box 3354, Riyadh, 11211, Saudi Arabia

**Keywords:** rectal brachytherapy, non-operative management of rectal cancer, conformed therapy for rectal cancer

## Abstract

**Simple Summary:**

With the introduction of better-quality imaging for tumor visualization and treatment planning, a new conformed radiation treatment was introduced with high-dose-rate endorectal brachytherapy (HDREBT). The advantage of this treatment is allowing for better sparing of normal tissues surrounding the tumor during treatment while delivering higher dose to the tumor. This diminishes the number and severity of side effects and results in more effective treatment. This manuscript summarizes two decades of technological evolution and progress in clinical studies to validate this treatment concept from the pre-operative setting to prevent tumor recurrence and, more recently, the introduction of the objective of cure without surgery; i.e., non-operative management (NOM) for patients with curable rectal cancer. HDREBT is a conformed radiation modality, shown to be safe and efficient both in the pre-operative setting and is presently being explored with interest in NOM in a multicenter study.

**Abstract:**

(1) Background: The introduction of total mesorectal excision (TME) for rectal cancer has led to improvement in local recurrence (LR) outcomes. Furthermore, the addition of preoperative external beam radiotherapy to TME reduces LR to less than 6%. As a trade-off to these gradual improvements in local therapies, the oncology community’s work is now focusing on mitigating treatment-related toxicities. In other words, if a small proportion of 4–6% of rectal cancer patients benefit from additional local therapy beyond TME, the burden of acute and long-term side effects must be considered with care. (2) Methods: With the introduction of better-quality imaging for tumor visualization and treatment planning, a new conformed radiation treatment was introduced with high-dose-rate endorectal brachytherapy. The treatment concept was tested in phase I and II studies: first in the pre-operative setting, and then as a boost after external beam radiation therapy, as a dose-escalation study, to achieve higher local tumor control. (3) Results: HDREBT is safe and effective in achieving a high tumor regression rate and was well tolerated in a phase II multicenter and two matched-pair studies. (4) Conclusions: HDREBT is a conformed radiation therapy that is safe and effective, and is presently explored in a phase III dose-escalation study in the NOM of patients with operable rectal cancer.

## 1. Introduction

The implementation of total mesorectal excision (TME) as the standard surgical approach for the treatment of rectal cancer has led to improvement in local recurrence (LR) rates, from a historical 25–30% to 6–12% [1,2,3,4]. Additionally, preoperative external beam radiation (EBRT) reduces this risk to less than 6% [5,6]. Given that local therapies are associated with excellent oncologic outcomes, the focus of contemporary innovation is shifting towards strategies to mitigate acute and delayed side effects. In 1998, At McGill University [7,8], a high-dose-rate endorectal brachytherapy (HDREBT) program was brought forth as an alternative downstaging modality in selected patients with resectable rectal cancer. The aim of this initiative was to reduce treatment-related normal tissue damage. This treatment option was possible with the introduction of two imaging modalities: first, magnetic resonant imaging (MRI), allowing for better tumor definition and staging; and second, the use of computed tomography (CT) images for treatment planning. The aim of this review is to summarize the history of this technological development and the evidence from clinical phase II trials from our center/collaborators, to support its potential use and limitations.

## 2. Pre-Operative HDREBT

### 2.1. Clinical and Dosimetric Aspects

When compared to EBRT, HDREBT delivers a higher dose rate of radiation with a quicker dose fall-off around the tumor target. This leads to improved sparing of normal tissues, such as the skin, prostate, bladder, and small bowel. The physical properties of the brachytherapy dose distribution inherent to the technique offer several advantages over external beam techniques. Within the tumor bed, a much larger radiation dose can be delivered, reducing the need for the sensitizing effect of chemotherapy (ChT). In addition, the tissues peripheral to the target volume are relatively spared. Lastly, since the treatment volume is smaller than that with EBRT, the treatment time can be shortened. Irradiation of the tumor and immediate perirectal nodes and intramesorectal deposits at a high dose with HBREBT, in order to achieve downstaging/downsizing, would, in turn, lead to negative circumferential mesorectal margins and might favor sphincter preservation surgery. Residual tumor cells and heavily irradiated tissues are removed at surgery. Consequently, it is reasonable to predict that the long-term toxicity on normal tissues will be low. The concept of testing this new modality in the preoperative setting is also appealing since the proof of its efficacy is possible with the availability of pathological specimens.

Based on MRI, Vuong et al. carefully selected T2–3 rectal cancer patients for a phase I–II study, mostly from the distal rectal segment without extra-mesorectal nodes. Tumor response rates were observed with pT0N0-1 as high as 32%, with an additional 38% of patients having small microscopic residual disease [8]. A local recurrence rate of 4.5% was observed over a median follow up time of 5 years and compared favorably to standard EBRT [9,10]. Acute proctitis was the only toxicity observed, with one percent of patients having a grade 3.

There is no randomized study to compare pre-operative HDREBT to EBRT. Nevertheless, two matched-pair studies [10,11] were conducted regarding acute post-operative and oncological outcomes. Hesselager et al. [11] compared the immediate post-operative outcome after EBRT with outcome after HDREBT. They used 318 patients treated with HDREBT followed by TME surgery 4–8 weeks later, matched with 318 patients treated with short-course radiotherapy (SCRT) and TME (RT+), and 318 patients treated with TME surgery alone (RT-) from the Swedish Rectal Cancer Registry. In this analysis, patient cohorts were matched for multiple characteristics, including age, gender, clinical stage and tumor height. The perioperative bleeding events were far less common in the HDREBT group in contrast with the SCRT and RT- groups. Indeed, the bleeding was 379.3 mL for HDREBT, 918.9 mL for SCRT (*p* < 0.0001), and 947.2 mL for RT- patients. The HDREBT patients after surgery experienced more cardiovascular complications (9.4%) than the SCRT group (3.1%, *p* = 0.002). Similarly, the RT- group experienced more cardiovascular complications than the SCRT group (*p* = 0.03). However, there was no difference in cardiovascular events between the HDREBT group (9.4%) and the RT-group (7.2%, *p* = 0.4) [11]. Septic complications were not significantly different between groups. Of note, more patients underwent repeat surgery after SCRT (14.2%) or RT (12.3%), when compared to HDREBT (4.1%, *p* < 0.0005). Macroscopic tumor clearance differed between groups, in favour of HDREBT (96.5%) over SCRT (83.3%) and RT (74.2%); the difference was statistically significant between HDREBT and RT (*p* = 0.03). The group which underwent HDREBT given pre-operatively experienced a complete tumour regression rate of 23.6%: we hypothesized that this may be a driving cause of our low rates of R2 (macroscopic positive margins) resection, and high proportions of R0 resection, in contrast with both Swedish groups. Additionally, it is possible that the conformed nature of HDREBT allowed for better normal tissue recovery in the postoperative phase, which may, in part, explain the favourable low rate of repeat surgery in this group. In summary, this analysis supported HDREBT as a safe treatment approach to downstage patients with rectal cancer, with limited perioperative bleeding and reoperation; however, the incidence of other complications resembled SCRT.

Breugom et al. [10] reported the long-term oncological outcomes (cancer-specific death, local recurrence, and overall survival) among 145 patients with clinical T3 rectal cancer treated with HDREBT (Canada), and 145 CT3 patients from The Netherlands treated either with TME alone, SCRT and TME, or long-course chemotherapy and external beam radiotherapy and TME. After a median follow-up of 6.6 years, the 5-year overall survival (OS) was 86.9% (95% CI: 80.1–91.6) for the Canadian cohort, and 70.9% (95% CI: 62.6–77.7) for The Netherlands cohort. Although the crude HR for OS was 0.62 (95% CI: 0.39–0.98; *p* = 0.040) between The Netherlands and Canadian centers, the adjusted HR for OS was 0.70 (95% CI: 0.39–1.26; *p* = 0.233). Among the 141 Canadian patients, 4.3% (95% CI: 0.9–7.5%) experienced a local recurrence and 10.6% (95% CI: 5.5–15.7) died of rectal cancer, compared to the 6.9% (95% CI: 2.8–11.0) local recurrence and 17.9% (95% CI: 11.7–24.2) rectal cancer deaths out of the 145 Dutch patients. There were no statistically significant differences in adjusted overall survival between the two countries. It was concluded that HDREBT is a safe alternative compared to the pre-operative strategy as used in an expert center in The Netherlands. Pre-operative HDREBT needs to be further investigated in a randomized controlled trial.

### 2.2. Locally Advanced, Stage III Rectal Cancer: The KIR Trial

Between 2010 and 2017, at a time when the role and timing of oxaliplatin-based chemotherapy were being investigated, Garant et al. conducted a multi-institutional randomized phase II clinical trial for patients with operable, locally advanced rectal cancer (NCT01274962) [12]. Patients were eligible if they had cT2–3, non-obstructing primary tumors, with at least one adverse radiographic finding on a baseline MRI, such as involved or close mesorectal fascia, cN+, or extramural venous invasion. A total of 180 eligible patients were randomly assigned (2:1) to two arms: either 6 cycles of FOLFOX prior to HDREBT and TME surgery followed by 6 cycles of adjuvant FOLFOX ChT(Arm A (AA)), or neoadjuvant HDREBT and TME with 12 cycles of adjuvant FOLFOX ChT(Arm B (AB)). The primary end point was compliance to ChT, defined as patients receiving at least 85% of the full-dose ChT prescribed at each cycle, for the first 6 cycles. Secondary end points were disease-free survival (DFS), ypT0N0, local control, and overall survival (OS).

All patients were randomly assigned to either AA (*n*  =  120; 84 patients were male (M), median age (MA) of 65 years) or AB (*n*  =  60; 35 patients were M, MA of 63.5 years). Subsequently, 175 of 180 patients completed HDRBT as planned (97.2%). In AA, 2 patients expired during ChT; additionally, 3 patients did not receive HDRBT after randomization due to progression under ChT (2 in AA) or personal preference (1 in AB) and received SCRT. The ypT0N0 for AA and AB were 36 patients (31%) and 17 patients (28%). Compliance was 80% on AA and 53% on AB (*p* = 0.0008). Levels of G3/G4 ChT toxicity were 35.8% in AA and 27.6% in AB, respectively. The median follow time was 48.5 months (IQR 33–72). The 5-year DFS was 72.3% with AA and 68.3% with AB (*p*  =  0.74). The 5-year OS for AA and AB were 83.8% and 82.2%, respectively (*p*  =  0.53). The 5-year local recurrence rate was 6.3% for AA and 5.8% for AB (*p* = 0.71).

The safety and improved compliance to neoadjuvant ChT was confirmed in this study using HDREBT as a neoadjuvant local therapy for rectal cancer. There is no statistical difference in the ypT0N0 rate, local recurrence, and DFS between the two arms, but favorable oncological outcomes have been observed. At the time of this reporting, pelvic nodal recurrence is seldomly isolated, asymptomatic, and associated with systemic failure. The safety and efficacy of a multi-centric application of HDREBT was established.

### 2.3. Technical Aspects

At the time when the clinical application of pre-operative HDREBT started, there was no CT-based commercially available treatment planning systems for brachytherapy. The target volume was initially imaged using ultrasound and MRI. Radio-opaque endorectal clips were then inserted under endoscopy in the patient to mark the proximal and distal tumor margins.

Patients were treated with preoperative high-dose-rate brachytherapy using the Novi Sad [13] endorectal applicator (Nucletron Corp., Columbia, MD, USA), with 8 catheters arranged around its circumference. The applicator (Figure 1) contained a balloon, which could be inflated so as to immobilize the device in the desired position in the rectum. Once the endorectal applicator was placed along the radio-opaque clips, patients underwent immediate CT simulation. The applicator catheters and the tumor contours were produced, and then projected onto digitally reconstructed radiographs (DRRs). In order to enhance the visualization of the bony landmarks, applicator, and clips, we used digitally composited radiographs (DCRs) and 3D renderings. For treatment planning, only dwell positions in catheters proximal to the tumor were selected. This source-positioning technique [13] allowed treatment of semi-circumferential lesions in a conformal manner. Following the source positioning, CT-based brachytherapy treatment planning was carried out to optimize the dose to the tumor. Dose distribution calculations were done using the in-house McGill Planning System (MPS), based on the AAPM TG-43 protocol [14]. Dose distributions were generated for a sequence of planes in arbitrary orientations and then superimposed onto two orthogonal DRRs.

A dose of 26 Gy in 4 daily fractions of 6.5 Gy was prescribed at the tumor radial margin. Prior to each treatment, AP and LAT daily check films were obtained with a mobile X-ray unit. The applicator position and orientation were verified by comparing these check films with the planning DRRs and adjustments were made if needed, which represented implementation of daily image guidance into brachytherapy treatments; i.e., image-guided brachytherapy (IGBT) [15]. Treatments were delivered using a Nucletron micro-selectron remote after-loading device that used a single 370 GBq (10 Ci) Ir-192 source at installation. The treatment workflow is presented in Figure 2.

In the year 2000, the first commercial treatment planning system (PLATO; Nucletron, Veenendaal, The Netherlands) became available. Clinical implementation of this treatment planning software allowed not only for more accurate 3D dose calculation based on CT images, but also provided an option (IPSA) for treatment plan optimization. The obtained dose distributions were highly conformal to the target volume, providing significant spearing of the surrounding critical structures (Figure 3).

In 2005, an intracavitary mould applicator of cylindrical shape (27 cm long and 2 cm in diameter) was introduced by Nucletron (Veenendaal, The Netherlands). The eight catheter channels included in the applicator provide an equal distribution of coverage around the entire circumference, in equal angular increments; furthermore, the central opening allows for the option of inserting an additional central catheter (Figure 4). The central lumen (8 mm in diameter) also allows for the insertion of a high Z material (lead, tungsten) shielding rod, to allow for even better spearing of healthy tissues contralateral (with respect to applicator) to the target volume [16]. Since the applicator material consists of silicon rubber, which is pliable, this device can easily be inserted and navigated within the rectum and sigmoid colon. When it comes to preoperative HDREBT, the addition of a ‘‘brachy-balloon’’ (CIVCO, Latex-Free Endocavity Balloon, 10e898 (BS3000)) prior to insertion provides the advantage of displacing the mucosa opposite to the tumor, when this balloon is inflated/oriented away from the target. With the introduction of the intracavitary mould applicator, we ceased to use the Novi Sad applicator.

Starting in 2009, a dedicated high-dose-rate brachytherapy suite with large-bore CT scanner became available to treat patients within the same room. In this setting, one decade after the creation of the IGBT technique, the workflow was replaced with daily adaptive CT-based HDREBT (Figure 5) [17].

## 3. Boost HDREBT

### 3.1. Clinical Aspects

In 2005, a phase II study was introduced for patients unfit or refusing surgery with rectal cancer with pelvic EBRT to a dose of 40 Gy in 16 fractions or 45–50 Gy in 25 fractions with concurrent 5-FU if eligible for ChT. This was followed by three weekly HDREBT boosts of 10 Gy to the residual clinical target volume, for a total of 30 Gy in 3 weekly fractions. Complete clinical response (cCR) was the primary endpoint. In our experience with 94 patients, data maintained in prospective database [18], we observed 86.4% cCR, a 12.8% proportion of regrowth, and a 72.8% local control rate at 2 years, with a 25.5% rate of late grade 3 bleeding. The vast majority of the patients accrued had significant comorbidities: this is reflected in the 2-year survival rate of 66.1%. These results compared favorably with those achieved with EBRT alone. In the same era, Wang et al. [19] reported on the Canadian experience of 271 elderly patients with localized rectal cancer treated with EBRT alone using various dose fractionations, with a mean dose of 40 Gy. With a median follow-up of 40 months, their rate of cCR was 30%, but the local recurrence was 78%, thus reflecting the limitations of EBRT.

Presently, non-operative management of rectal cancer is of major interest to patients and the oncology community as the population is getting older and the incidence of colorectal cancer is increasing. Hall [20] analyzed individual data of 3298 rectal cancer patients (treated between 2000 and 2013) using a pooled database of cancer registries from more than 150 US hospitals. Eligible patients were treated with ChT-EBRT followed by surgery and had complete data on treatment. We aimed to quantify the predictive value of the independent variables to achieve a pathologic complete response (pCR) using multivariable logistic regression. The most significant factor in achieving pCR was a high radiation dose, with 10.9% at the 45 Gy dose to 18.8% at 54 Gy in patients with stage II and III cancer. A subsequent meta-analysis from Sanghera et al. [21] supported this finding. Additionally, an additional investigation by Appelt et al. [22] noted a direct association between the radiation prescribed dose and tumor regression, with an optimal regression at biologically equivalent doses above 92 Gy. However, this leads to clinical impasse, since the tolerance of the normal rectal mucosa is in the range of 53–75 Gy, depending on the exposed mucosal volume [23,24]. In this context, rectal brachytherapy is becoming the most appealing tool to allow for cCR in the NOM of rectal cancer.

Two other centers also reported on the use of HDREBT as a boost technique for rectal cancer patients [25,26]. There are some technical differences between non-adaptive brachytherapy [25] and our approach. In the former, the boost volume is determined once after EBRT, whereas in our technique, it is being assessed at multiple time points. Second, the radiation prescription volume differed between the two studies: in non-adaptive brachytherapy, a 2 cm depth of prescription was allowed, which could lead to 600% of the prescription dose being administered to rectal mucosa. In our technique, the double balloon system minimizes the amount of intratumoral dose gradient by deforming and flattening the tumor target to a thickness of 1 cm or less. The resulting volume and dose prescription depth differences could explain the difference in late grade ≥3 proctitis from our series with 19.2% [18] and 40% for the HERBERT trial [25]. The Danish study [26] used a lower brachytherapy dose and had 6% of grade 3 bleeding with a reported 300% mucosal dose. Indeed, the total boost dose of 5 Gy certainly accounts for it; however, all patients with persistent tumor underwent salvage surgery. These dose differences and late toxicity rates show that treatment volume and mucosal dose are the critical factors for rectal toxicity, in accordance with randomized prostate cancer trials [24], with caveats related to possible differences in patient populations.

We are presently conducting a phase III randomized multicentre study to validate the value of HDREBT in patients with operable stage II rectal cancer (NCT03051464). The preliminary interim toxicity analysis on the first 40 patients [27] showed favourable toxicity post-operative data and potential benefits from HDREBT boost in the NOM of rectal cancer. A similar study design was conducted by Mynt et al. [28] using contact brachytherapy and the results are to be presented at an ASCO meeting, showing the definitive benefits of dose escalation.

### 3.2. Technical Aspects

Since 2005, with the introduction of the intracavitary mould applicator, inoperable patients were given a boost dose to the tumor site, after EBRT, in accordance to the workflow presented in Figure 6.

Poon et al. [29] provided Monte Carlo simulation results that allow for implementation of the tungsten shielding technique, with the objectives of limiting the dose to contralateral healthy tissues and mitigating the amplitude of long-term toxicities. Following the results from this initial study, the group at McGill established an internal guideline to employ shielding (0.7 cm diameter tungsten rod; slightly smaller than the applicator central lumen diameter to allow for a smooth fit) in select scenarios, only if three consecutive applicators or less are required to cover the CTV. In the setting of the boost technique, the high dose gradient along the target is softened using an additional ipsilateral balloon, which flattens the tumor. This ipsilateral balloon principle is illustrated in Figure 7.

To summarize, using a lubricant gel, two balloons were integrated along the applicator before insertion: this approach optimizes the dose conformity of the HDREBT boost technique. It remains of utmost importance to remember which balloon is placed along the tumor versus on the contralateral side of the rectal lumen. Indeed, after the insertion of the applicator, the ipsilateral balloon is inflated with 30 cm^3^ water, whereas the contralateral balloon is inflated with 50 cm^3^ water. After the applicator is immobilized and the balloons are inflated, the CT scan and treatment planning steps follow. Upon confirmation of a satisfactory plan, the final central intracavity shield is placed before the start of the treatment.

## 4. Conclusions

The introduction of modern diagnostic imaging with MRI and a precise radiation treatment planning system with a CT simulator led to innovation of conformed radiation modality with HDREBT. This innovative treatment modality was tested by pre-operative studies where efficacy was validated by pathological specimen and oncological outcomes. Two matched-pair studies showed the safety of pre-operative HDREBT and favourable but not statistically significant oncological results. Similar trends were observed in a randomized multicenter phase II study, showing the safety and efficacy of the pre-operative setting in patients with more advanced disease at risks for metastasis; however, there are no randomized study comparing HDREBT to EBRT. In the era where NOM is of major interest to patients, HDREBT might become a major addition to the management of rectal cancer. HDRBT treatment units exist in most radiation oncology centers, unlike intracavitary contact X-ray mobile units. The intracavitary mould applicator is commercially available. The only step necessary to initiate the technique is clinical training. The definitive role of HDRBT is presently tested in the phase III randomized Morpheus study. Interim results were recently published [30] and the study is actively recruiting. The concept of dose escalation was also tested in the OPERA trial, the results of which were reported at the ASCO meeting and the final paper has been submitted for publication.

Despite its clear clinical and long-term advantages for patients treated with HDREBT, the technique has some disadvantages. In the pre-operative setting, for patients with tumors extending deeper than 2.5 cm, the dose distribution would not be deemed optimal. The technique is also labour intensive and requires more comprehensive resources when compared to EBRT. 

To reach the level of standard of care, a phase III clinical trial is needed, and our group is working on organizing it for quite some time now. At this very moment, use of central shielding is also tailored empirically, resulting in the use of three channels only. We are working on developing the Monte Carlo-based dose calculation and optimization that would allow us not only to use more than three channels but also to better optimize the planning process.

## Figures and Tables

**Figure 1 cancers-14-04846-f001:**
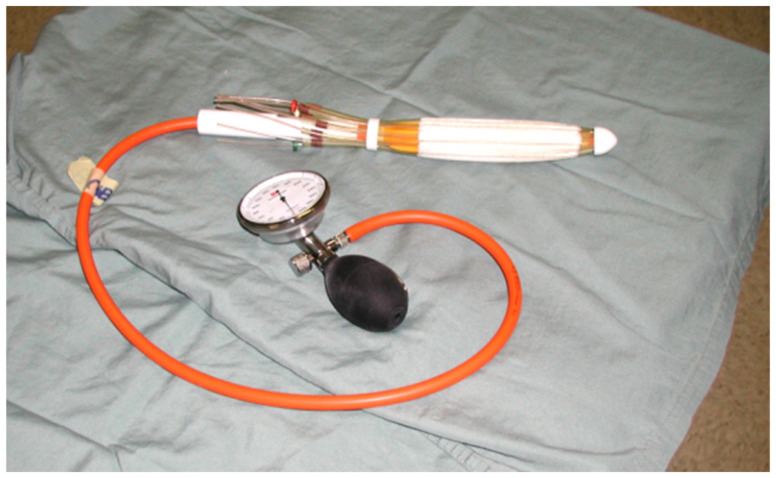
Novi Sad [13] endorectal applicator (Nucletron Corp., Columbia, MD, USA).

**Figure 2 cancers-14-04846-f002:**
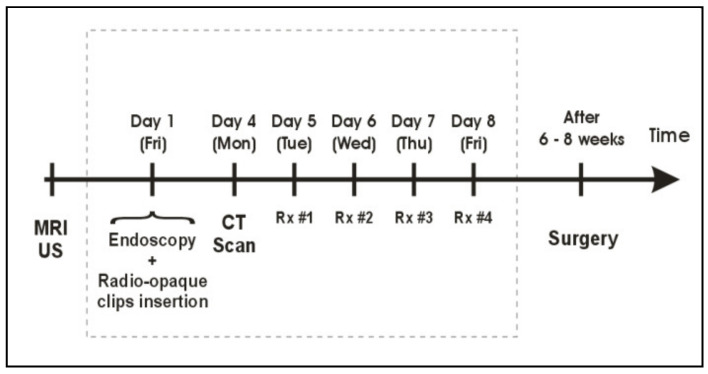
Treatment workflow for pre-operative HDREBT.

**Figure 3 cancers-14-04846-f003:**
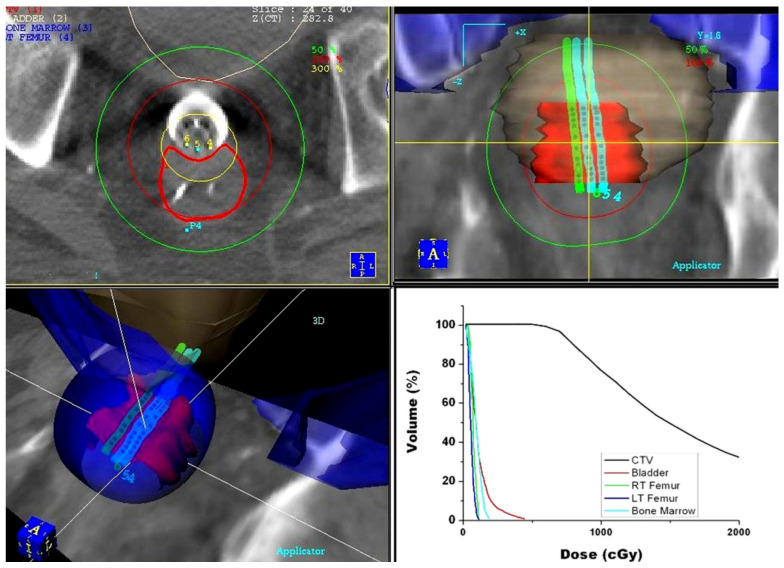
Dose distribution for pre-operative endorectal brachytherapy treatment obtained using the PLATO treatment planning system: axial (top left), coronal (top right), and 3D (bottom left) view of the dose distribution. Dose–volume histograms (bottom right) of the target volume and surrounding critical structures.

**Figure 4 cancers-14-04846-f004:**
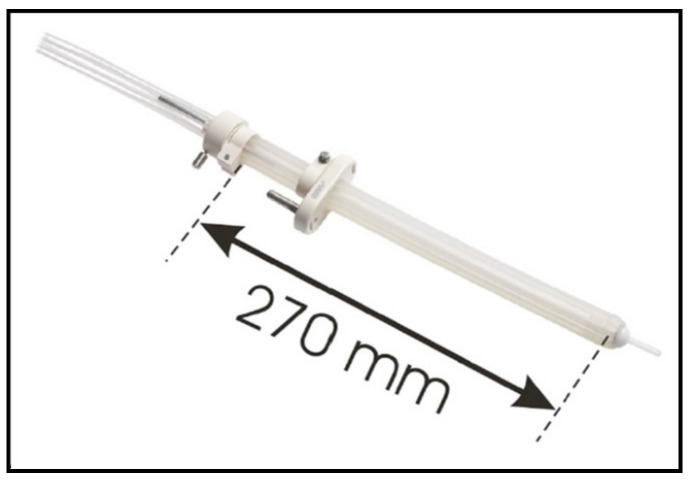
Intracavitary mould applicator.

**Figure 5 cancers-14-04846-f005:**
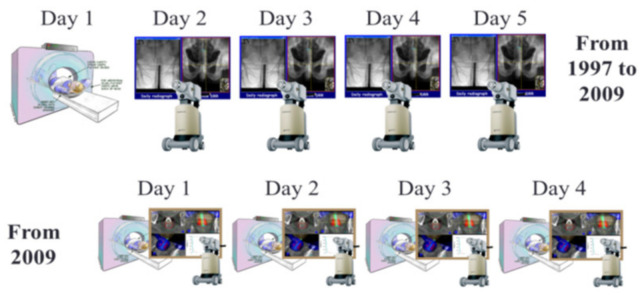
Adaptive image-guided HDREBT: top—workflow for image-guided brachytherapy (IGBT); bottom—workflow for daily adaptive brachytherapy.

**Figure 6 cancers-14-04846-f006:**
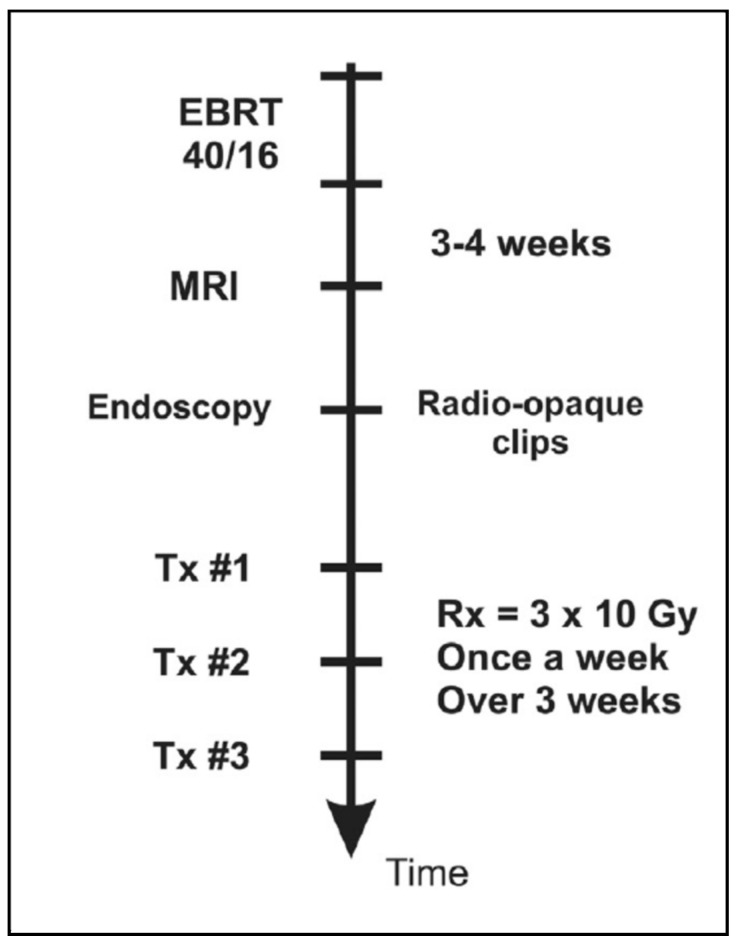
Workflow of brachytherapy boost for inoperable rectal cancer patients consisting of three treatment fractions given as one fraction per week.

**Figure 7 cancers-14-04846-f007:**
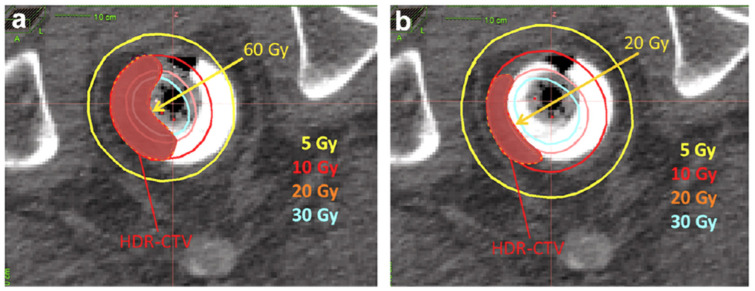
Role of the ipsilateral balloon on lowering the mucosa dose: (**a**) dose distribution without ipsilateral balloon—mucosa receives 60 Gy; (**b**) dose distribution with ipsilateral balloon—mucosa receives 20 Gy.

## Data Availability

Not applicable.

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
