# Peer review of "Image-Guided Brachytherapy for Rectal Cancer: Reviewing the Past Two Decades of Clinical Investigation"

_cancers, 2022, doi:10.3390/cancers14194846_

Round 1

Reviewer 1 Report

In this current manuscript, Vuong et al summarized the improvement made in radiation therapy in rectum cancer in last two decades. Their accumulated evidences show the chronological aspect of improvement of Highly Targeted Radiation Therapy from the conventional radiotherapy. HDREBT was found to be a more efficient technique in terms of patient survival, recurrence and mitigating adjacent host tissue damage. They author has done a thorough review of literate study and summarized the sequential events in the improvement of this targeted radiation therapy technique in terms of clinical aspects and technical aspects.  Over all this is interesting manuscript to read, and give a border view of this field.  The manuscript written nicely and summarized the major studies done in this field, conclusion from the other studied were done perfectly.  This manuscript will probably get future attention from a wide range of oncology surgeon and basic cancer researcher.  

Comments- No major comments.

Minor-

1.      Author can separately add a paragraph for any limitation of this high beam radiation technique.

2.      Author suggestion for future improvement of the technique in a paragraph will be a good addition.

3.      A small legend in the figures will increase the understating of those figure.

Author Response

Comment 1: Author can separately add a paragraph for any limitation of this high beam radiation technique.

Reply: In the Conclusion section at the end, one paragraph before the last one, we added paragraph:

“Despite its clear clinical and long term advantages for patients treated with HDREBT, technique has some disadvantages. In the pre-operative setting for patients with tumors extending deeper than 2.5 cm the dose distribution would not be deemed optimal.  Also, the technique is labour intensive and requires more comprehensive resources when compared to EBRT.”

Comment 2: Author suggestion for future improvement of the technique in a paragraph will be a good addition.

Reply: In the Conclusion section at the end, we added paragraph:

“To reach the level of standard of care, a phase III clinical trial is needed and our group is working on organizing it for quite some time now. Also, at this very moment, use of central shielding is tailored empirically resulting in use of three channels only. We are working on developing the Monte Carlo based dose calculation and optimization that would allow us not only to use more than 3 channels, but also to better optimize the planning process.”

Comment 3: A small legend in the figures will increase the understating of those figure.

Reply: We added explanatory text to almost all figures in the manuscript.

Reviewer 2 Report

I congratulate the authors of the "Highly Targeted Radiation Therapy for Rectal Cancer: Reviewing the Past Two Decades of Clinical Investigation" paper. They present a very novel topic under the light of growing interest in NOM for rectal cancer.

They describe properly the importance of why this article is important to any readership. The references are sound and proper introduce during the text.

I have a few recommendations:

I recommend trying to add a statement of which is the main aim of the review and also a brief paragraph of how data was recollected (i.e. a systematic review of the current evidence was performed or the most relevant articles about this topic were included). 

An appropriate level of evidence for the current data available is presented. I recommend to add a paragraph of how the results of this technique is similar to other types of rectal brachytherapy as well (i.e Herbert Study, Int J Radiat Oncol Biol Phys 2017;98:908-917.)

I recommend to state that the Phase 2 study that is presented (REF 18) correspond truly to prospective maintained database. The original paper stated: “In 2011, the study was amended as a multicenter registry for patient unfit for standard management”. 

In the conclusions, I recommend to state that HDREBT may be a major addition to the management of rectal cancer (rather than "will be"), but further trials are needed to safety add it as part of the standard of treatment for patients with rectal cancer.

Currently, I recommend making this minors changes to considerer the present paper to be accepted.

Author Response

Replies to Reviewer 2

Comment 1: I recommend trying to add a statement of which is the main aim of the review and also a brief paragraph of how data was recollected (i.e. a systematic review of the current evidence was performed or the most relevant articles about this topic were included).

Reply: To address this suggestion, at the end of Introduction section, we added a sentence:” The aim of this review is to summarize history of this technology development and clinical phase II trials from our center/collaborators and evidence to support its potential use and limitations.”

Comment 2: An appropriate level of evidence for the current data available is presented. I recommend to add a paragraph of how the results of this technique is similar to other types of rectal brachytherapy as well (i.e Herbert Study, Int J Radiat Oncol Biol Phys 2017;98:908-917.)

Reply: To address this suggestion, we added a paragraph:

“Two other centers were also reported on the use of HDREBT as a boost technique for rectal cancer patients [25, 26]. There are some technical differences between non-adaptive brachytherapy [25] and our approach. In the former, the boost volume is determined once after EBRT, whereas in our technique, it is being assessed at multiple time points. Second, the radiation prescription volume differed between the two studies: in non-adaptive brachytherapy, a 2 cm depth of prescription was allowed, which could lead to 600% of the dose being administered to rectal mucosa. In our technique, the double balloon system minimizes the amount of intratumoral dose gradient by deforming and flattening the tumor target to a thickness of 1 cm or less. The resulting volume and dose prescription depth differences could explain the difference in late grade ≥3 proctitis from our series with 19.2% [18] and 40% for the HERBERT trial [25]. The Danish study [26] used a lower brachytherapy dose, had 6% of grade 3 bleeding with a reported 300% mucosal dose. Indeed, the total boost dose of 5 Gy certainly accounts for it; however, all patients with persistent tumor underwent salvage surgery. These dose differences and late toxicity rates show that treatment volume and mucosal dose are the critical factors for rectal toxicity, in accordance with randomized prostate cancer trials [24] with caveats related to possible differences in patient populations.”

Comment 3: I recommend to state that the Phase 2 study that is presented (REF 18) correspond truly to prospective maintained database. The original paper stated: “In 2011, the study was amended as a multicenter registry for patient unfit for standard management”.

Reply:  in the sentence “In our experience with 94 patients,” we added statement: “… data maintained in prospective database [18], …”

Reviewer 3 Report

The authors summarize the past two decades of technological evolution and clinical investigation of a new targeted radiation treatment for rectal cancer with high dose rate endorectal brachitherapy (HDREBT). This new treatment modality involved patients in the preoperative setting and, more recently, in the organ preservation approach with non-operative management for selected cases with curable disease.

The authors report a detailed review on the available evidences of the feasibility and efficacy of preoperative HDREBT including the matched pair studies comparing the oncologic outcome and perioperative morbidity of preop HDREBT and TME surgery versus conventional preop short course RT and TME or surgery alone (data from Swedish Rectal Cancer Registry). Data on comparison of long term oncological outcomes between Canadian and Dutch expert centers, and randomized phase II trial comparing preop HDREBT  with adjuvant or neoadjuvant chemotherapy are also well presented and discussed. Data from these studies represent a major strength of the paper.

In addition, the authors reported a detailed technical aspects on HDREBT planning and treatment procedure, and the evolving technological solutions from  the more traditional Novi Sad endorectal applicator using the McGill Planning System for dose distribution calculation, to the more recent Intracavitary Mold Applicator allowing a better dose conformality optimization  and daily adaptive CT-based HDREBT.

Such precise HDREBT planning system allowed safe dose escalation programs after external RT with HDREBT boost for patients unfit or refusing surgery. The authors well reported and discussed the initial phase II studies in non-operative approach in surgical unfit patients and the ongoing clinical investigations with HDREBT  in the emerging interest of non-operative management (NOM) in selected patients with operable stage II-III rectal cancer.

In their conclusions, the authors confirm the investigational interest of HDREBT in NOM of rectal cancer, and its easily accessible treatment technique without the need of dedicated treatment unit. The expertise can be achieved in a reasonable time..

The paper is of great interest in reviewing clinical investigations of HDREBT, either in preopertative setting than in exploring of NOM of rectal cancer, combined with technological evolution of this new treatment modality.

I would suggest to the authors some comments more on the accessibility of treatment technique; the authors reported that dedicated treatment units/facilities are not required while a dedicated intracavitary mold applicator with different options for conformal dose distribution are described (Fig.4)

Other minor comments.

Line 59 ... review references, probably 1-4 instead 5,6

Line 60   add references 5,6 after reduce risk less than 6%

Line 75  I suggest to add a reference to fig 1

Lines 97 to 100 this sentence is not clear; probably the full stop after HDREBT could be removed

Line 109 and line 117 .. repeated surgery shoul be specified (complications? recurrent/persistent disease?)

Line 66  I suggest to specify Technical Aspects (traditional applicator or Novi Sad applicator) to distinguish from Technical Aspects (Intracavitary mold applicator) in line271

Author Response

Replies to Reviewer 3

Comment 1: I would suggest to the authors some comments more on the accessibility of treatment technique; the authors reported that dedicated treatment units/facilities are not required while a dedicated intracavitary mould applicator with different options for conformal dose distribution are described (Fig.4).

Reply: In the Conclusion section at the end of the first paragraph, line 347, we added a sentence:

“HDRBT treatment units exist in most Radiation Oncology centers unlike the Intracavitary contact X Ray mobile unit. The intracavitary mould applicator is commercially available. The only step necessary to the initiation of the technique is clinical training.”

Comment 2 Line 59 ... review references, probably 1-4 instead 5,6

Reply: Correct – we changed it accordingly.

Comment 3 Line 60 add references 5,6 after reduce risk less than 6%

Reply: Correct – we changed it accordingly.

Comment 4: Line 75 I suggest to add a reference to fig 1.

Reply: Unfortunately, this figure was never published.

Comment 5: Lines 97 to 100 this sentence is not clear; probably the full stop after HDREBT could be removed.

Reply: In order to make the sentence clear, after “… HDRBT.” we added, at line 105: “They used …”

Comment 6: Line 109 and line 117 .. repeated surgery should be specified(complications? recurrent/persistent disease?).

Reply: We added “… after surgery …” on line 112 (old line 109), and on line 121 (old line 117), we added after the word HDREBT “… given pre-operatively…”.

Comment 7: Line 66 I suggest to specify Technical Aspects (traditionalapplicator or Novi Sad applicator) to distinguish from TechnicalAspects (Intracavitary mold applicator) in line271.

Reply: To make distinction in time between Novi Sad and intracavitary mould applicator, at line 233 we added the following text: “With the introduction of the intracavitary mould applicator we ceased to use the Novi Sad applicator.”